# Application of Novel Technologies in Cardiac Electrotherapy to Prevent Complications

**DOI:** 10.3390/diagnostics13091584

**Published:** 2023-04-28

**Authors:** Szymon Budrejko, Maciej Kempa, Justyna Rohun, Ludmiła Daniłowicz-Szymanowicz, Agnieszka Zienciuk-Krajka, Anna Faran, Grzegorz Raczak

**Affiliations:** Department of Cardiology and Electrotherapy, Medical University of Gdansk, 80-214 Gdansk, Poland; budrejko@gumed.edu.pl (S.B.); kempa@gumed.edu.pl (M.K.); justynarohun@gumed.edu.pl (J.R.); agzien@gumed.edu.pl (A.Z.-K.); anfar@gumed.edu.pl (A.F.); gracz@gumed.edu.pl (G.R.)

**Keywords:** implantable cardioverter-defibrillator, subcutaneous implantable cardioverter-defibrillator, transvenous lead extraction, infective endocarditis, cardiac pacing, complications, pulmonary valve insufficiency

## Abstract

(1) Background: Cardiac electrotherapy is developing quickly, which implies that it will face a higher number of complications, with cardiac device-related infective endocarditis (CDRIE) being the most frequent, but not the only one. (2) Methods: This is a retrospective case study followed by a literature review, which presents a patient with a rare but dangerous complication of electrotherapy, which could have been prevented if modern technology had been used. (3) Results: A 34-year-old female was admitted with suspicion of CDRIE based on an unclear echocardiographic presentation. However, with no signs of infection, that diagnosis was not confirmed, though an endocardial implantable cardioverter-defibrillator (ICD) lead was found folded into the pulmonary trunk. The final treatment included transvenous lead extraction (TLE) and subcutaneous ICD (S-ICD) implantation. (4) Conclusions: With the increasing number of implantations of cardiac electronic devices and their consequences, a high index of suspicion among clinicians is required. The entity of the clinical picture must be thoroughly considered, and various diagnostic tools should be applied. Lead dislocation into the pulmonary trunk is an extremely rare complication. Our findings align with the available literature data, where asymptomatic cases are usually effectively treated with TLE. Modern technologies, such as S-ICD, can effectively prevent lead-related problems and are indicated in young patients necessitating long-term ICD therapy.

## 1. Introduction

Cardiac implantable electronic devices (CIEDs) have been used to treat cardiac rhythm abnormalities for over 60 years. Two influential groups among those devices are pacemakers (used to treat bradycardia) and implantable cardioverter-defibrillators (ICDs, intended for tachyarrhythmia treatment and sudden cardiac death (SCD) prevention). Since the first in-human implantation of CEIDs [1,2], the indications, technical capabilities of the devices, and guidelines for this type of cardiac treatment have evolved substantially [3,4].

However, as the number of procedures has increased and is still increasing exponentially [5], the number of complications necessitating specialist attention has unavoidably risen. Herein, infective complications are of significant concern [6,7,8,9]. According to Prutkin et al. [6], the long-term rate of total infections over three years after implantations was 1.4%, 1.5%, and 2.0% for single, dual, and biventricular ICDs, respectively [6]. Other authors have presented similar results [7,8,9]. Cardiac device-related infective endocarditis (CDRIE), the most frequent and dangerous infection complication, is of significant clinical importance [9,10,11,12]. Nonetheless, other complications, such as broken or dislocated leads, often with additional thrombus formation, are also significant, and their management may be demanding [13,14,15,16,17,18]. Lead extraction is often an essential part of the treatment; however, additional long-standing therapy with antibiotics is recommended for CDRIE, but not for other non-infective complications of electrotherapy [19]. Therefore, determining the appropriate diagnosis is crucial. It is noteworthy that the discovery and introduction of subcutaneous ICDs (S-ICD) into clinical practice was a milestone in cardiac electrotherapy and the prevention of its complications. Lead-related problems or a patient’s young age necessitating an ICD long-term are listed as indications for its implantation [3]. 

Nowadays, echocardiography is a first-line imaging technique that visualizes moving structures connected with electrodes, which could result from CDRIE. However, in some patients, thrombus formation due to lead damage or dislocation can result in a similar echocardiographic picture, making the proper diagnosis challenging. Precise clinical collection of the patient’s history, laboratory analysis, and possibly tests using other diagnostic tools should be performed to inform the final decision. 

The current paper aims to present a case of extremely rare ICD lead dislocation into the pulmonary trunk with thrombus formation, mimicking the CDRIE process, in a young female patient. Further on, a literature review of similar findings is provided.

## 2. History of Presentation

A 34-year-old female with a diagnosis of hypertrophic cardiomyopathy (HCM), and an ICD implantation performed in 2010 for the primary prevention of SCD [7], was admitted to our department in September 2021 with suspicion of CDRIE based on TTE echocardiography findings; the right ventricular (RV) lead was thickening (up to 10 mm), and mobile structures (approximately 8 mm in length) attached to the RV lead within the right atrium were visualized (Appendix A).

### 2.1. Previous History

Since the implantation, the patient underwent regular follow-ups in our outpatient clinic. The ICD pacing and sensing parameters were checked repeatedly and were continuously correct (Table 1).

Pharmacological treatment was based on beta-blockers (the last reported was slow-release metoprolol (100 mg, once daily)). In the previous TTE HCM with preserved systolic function (left ventricular (LV) ejection fraction of 70%), neither systolic anterior motion (SAM) nor obstruction in the LV outflow tract were described, but an increased mid-ventricular systolic pressure gradient (from 30 at rest to 60 mmHg during Valsalva maneuver) was. The patient reported no significant complaints until 2019, but in September 2019, she started suffering from worsening exercise capacity. Repeated Holter-ECG and an echocardiographic study did not reveal any pathology.

### 2.2. Presented Hospitalization

On admission, the patient was hemodynamically stable and consistently denied having a fever or any other symptoms of infection. Additionally, there was no other inflammatory state or invasive procedure during the preceding six months. Moreover, the patient was vaccinated against SARS-CoV-2 with a full dose. A systolic murmur in the area of auscultation of the pulmonary artery (PA) valve (II left intercostal space) was found in a physical examination. In the laboratory analysis, no signs of infection were noted (C-reactive protein < 0.40 mg/L, procalcitonin: 0.01 ng/mL, d-dimer: 302 microgram/L, hemoglobin: 13.1 g/L, platelet count: 292 G/L, red blood cells: 4.57 T/L, white blood cells: 9.11 G/L—Table 2).

As the blood cultures (aerobic and anaerobic) were repeatedly negative, the empirical treatment of suspected infective endocarditis was not implemented. In TTE, in addition to a thickened RV lead with attached mobile structures, the features of the lead dislocation into the PA were revealed via an echo of the lead’s presence with intensive doppler color of pulmonary regurgitation, visualized in PA (Figure 1, Appendix A).

All TTE findings were confirmed via transesophageal echocardiography (TEE). Moreover, the chest X-ray (CXR) confirmed the dislocation of the ICD lead into the PA (Figure 2). This image was compared with the one obtained immediately after the implantation of the ICD in 2010, where the typical location of the lead was presented (Figure 3). In Figure 3, the length and loop of the lead in the pocket under the ICD are smaller than in Figure 2, indicating that the lead might have been drawn further into the vein compared with its initial course after implantation.

The patient qualified for ICD lead extraction, and the indications for ICD implantation were re-evaluated. According to the current guidelines [3], the calculated HCM risk score for the patient was equal to 9.36%, which was a class IIa indication for ICD implantation. The patient’s preference was to have the ICD re-implanted, and this notion mainly seemed related to the history of sudden cardiac death in her close family. Due to the young age of the patient, the long-term need for ICD therapy, and the lack of necessity for stimulation or anti-tachycardia pacing (ATP) functions, the novel technique of implanting an S-ICD following the transvenous removal of the current ICD system was considered to be the best solution in that particular case. Electrocardiography (ECG) screening for S-ICD was positive in all three leads, making the patient eligible for such a treatment. The transvenous ICD lead was removed using a TightRail rotational mechanical sheath (Philips Healthcare, the Netherlands). No intra-operative complications occurred, and the S-ICD (Emblem MRI, Boston Scientific) was implanted. The removed electrode was thoroughly visually examined by the operator for potential changes and/or damage—no alteration was found. Venography could have added some important information, particularly before transvenous re-implantation. However, in our case, we did not perform this, as the procedure was S-ICD re-implantation and the operators did not consider venography necessary; additionally, we minimalized the dosage of radiation to the patient. The CXR images with the new S-ICD are presented in Figure 4.

The culture of the extracted transvenous lead was negative. No signs or symptoms of infection were observed after surgery. Moderate pulmonary insufficiency was present in the postoperative TTE (Appendix A) without RV overload or dysfunction features. In a follow-up three months after the surgery, no infective complications or any features of RV overload were found in the patient, despite persistent pulmonary insufficiency.

## 3. Discussion

The main finding of our study is that physicians should be aware of implantable cardiac electronic device complications and modern technologies’ ability to prevent them. Although infective endocarditis is the most frequent, some rare but dangerous complications must be considered.

Our patient had mobile structures, believed to be bacterial vegetations, of up to 8 mm in length, attached to the lead on its course through the right atrium. However, the medical history, the risk factors, and all the other investigation results spoke against infection in that case. Precise echocardiography and simple CXR allowed for finalization of the appropriate diagnosis. This led to pulmonary valve insufficiency and caused mild heart failure (HF) symptoms. Such findings could explain the TTE and TEE images (which were consistent with pulmonary regurgitation), auscultation (systolic murmur on the PA), the patient’s symptoms (worsening exercise capacity), and possibly also the formation of thrombi on the lead‘s body due to its atypical course, and the resulting turbulent blood flow. 

Concerning cardiac implantable electronic devices, pacing lead displacement is a relatively frequent complication. According to a recent meta-analysis of the incidence of lead dislodgement by Wang MS et al., the incidence of this phenomenon in the included studies ranged from 1% to 2.69% [20]. Furthermore, it is among the most common reasons for re-intervention following device placement. A prospective registry of at least one year of follow-ups regarding re-intervention in patients with CIEDs by Ghani et al. revealed that lead dislodgment was present in 66% of cases [21]. However, despite its importance, studies on the incidence and management of lead displacement are scarce. Thus far, late lead dislodgment into the pulmonary artery (LDPA) has only been found in a few reports, and is usually due to lead fracture and intrusion of the distal fragment into the venous system [15,17,22,23,24]. Casuistically, macro-dislocation due to unconscious manipulation of the pulse generator around its axis by the patient, known as twiddling the pacemaker, was described [25]. This is a particularly difficult complication to predict before CIED implantation and includes a group of three syndromes, called Twiddler’s, reel, and ratchet syndromes. Twiddler’s and reel syndromes are defined as the lead retracting as a result of the impulse generator rotating around its long and horizontal axes, respectively. Moreover, in Twiddler’s syndrome, spiral-shaped twisting of the lead around itself is seen, whereas, in reel and ratchet syndromes, the lead is observed to be circled around the device. It is noteworthy that in the literature, Reverse Ratchet syndrome, consisting of anti-clockwise rotation of the pacemaker around its transverse axis due to arm movements and loose lead fixation, leading to alteration in the lead position, can also be found [25]. The primary literature findings on the subject are presented in Appendix A. In our study, no pulse generator location change was found, nor was the twisting of electrodes around the device, which indicates that the case cannot be attributed to any of the above-mentioned twiddling pacemaker syndromes. The defibrillating ICD lead was drawn into the venous system from the pocket, possibly due to inadequate fixation on the anchoring sleeve. The suture was probably not tight enough, so the electrode retracted from the right ventricle, causing its dislocation, and its excessive loop was folded into the pulmonary trunk.

The largest study assessing LDPA in patients with CIEDs was conducted by Polewczyk et al., and revealed an estimated incidence of LDPA of 1.07% [13]. Other studies mainly include case reports. The population with LDPA varied from children [24] to the elderly [14,17,18,25]. The symptoms of such a dislocated lead are not specific and include, for example, fever, cyanosis, and decreased exercise capacity, or may not be present at all [15,16,24]. Sometimes, dislodgement, which occurs through the creation of turbulent blood flow, leads to increased thrombotic status and can result in thrombi around the lead, which may occasionally cause a pulmonary embolism [26,27]. As a routine CXR usually allows doctors to confirm dislocation with a very high probability [14,15,17,24,25,28,29], multimodal imaging is remarkably beneficial in the diagnostic pathway [30]. Primarily the CXR should be performed early, even if it is not considered a first-line study, i.e., when suspecting an infection such as CRIE due to an unclear TTE/TEE image. 

Regarding treatment methods for LDPA, TLE is likely to be efficient and safe. According to the available literature, it can be stated that after the transvenous removal of the lead using modified techniques, no complications after the procedure are usually observed [31,32]. TLE and targeted treatment for other conditions have been performed with high efficacy [13,14,15,16,18,23,24,25,32]. Sometimes, if the lead is left in the pulmonary circulation, it might remain clinically silent [18,33].

With the support of all the presented data, we speculate that in the reported case, lead intrusion into the venous system with an excessive loop led to its folding into the pulmonary trunk, which caused pulmonary regurgitation and mild symptoms of HF. By creating turbulent blood flow, the condition increased the possibility of thrombosis and the formation of thrombi attached to the lead body. The extensive clinical workup might have been limited if a CXR had been performed earlier in the process. However, the initial findings of structures presumed to be vegetation set the diagnostic algorithm toward suspected infection. Moreover, if a remote monitoring system in Poland was funded and routinely used in patients with CIEDs, some alterations in transmission signals would have prompted the clinicians to quickly initiate the diagnostics pathway. Eventually, transvenous extraction of the system decreased the risk of thromboembolic events and removed the foreign body from the lumen of the PA valve. The S-ICD implantation was considered an optimal solution to preventing SCD in the patient. This novel technique is highly beneficial in patients with difficult venous access after removal of the transvenous system due to infective complications, or in young individuals with a long-term need for ICD therapy, as was the case with our patient [3]. Due to the unavailability of pacing function and ATP therapy, the device is not routinely used, and the decision to use it is based on an individual approach. In our patient, the functions mentioned above were not required, and the risk-to-benefit profile was higher for applying treatment with S-ICD. 

The presented case report comes from the high-volume arrhythmic center, one of the leading centers in electrotherapy and the management of its complications. Since the first implantation of an S-ICD in Poland in October 2014, the procedure has been successfully introduced to clinical practice. No early or late complications have been observed so far [34,35,36]. The experience of our department is similar. If this modern technique had been available during the primary implantation of the cardiac device in our patient in 2010, the presented complications that arose from the dislocated lead would not have occurred.

## 4. Conclusions

Evaluating patients with cardiac implantable electronic devices and related complications may be challenging and require a high index of suspicion. A dislocated intracardiac lead may cause blood flow disturbances, possibly leading to lead thrombosis and needing clear differentiation with vegetations. Precise clinical evaluation and multi-modality imaging, including simple tests such as a chest X-ray, can be of great value for providing a final diagnosis. Novel cardiac electrotherapy, such as S-ICD implantation, seems efficient in preventing potential complications and should always be considered in individual cases, such as in young patients.

## Figures and Tables

**Figure 1 diagnostics-13-01584-f001:**
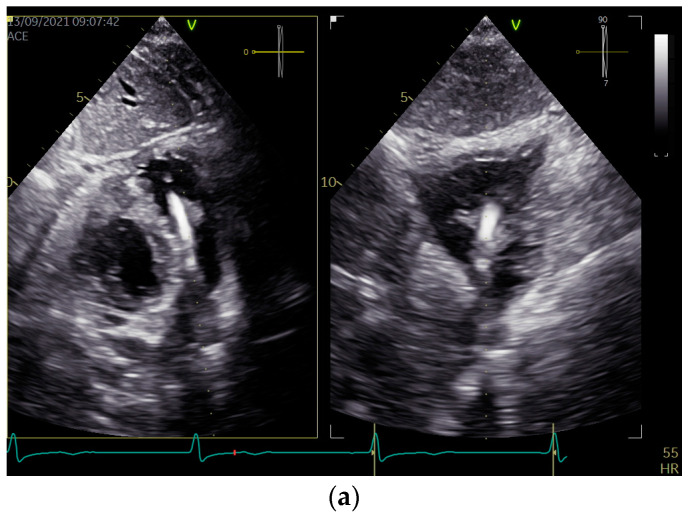
Two-dimensional transthoracic echocardiographic views show the implantable cardioverter-defibrillator (ICD) lead located in the pulmonary outflow tract (**a**), causing pulmonary insufficiency (**b**). The examination was performed using a GE VIVID E95 machine (GE ultrasound System, Horten, Norway) equipped with an active-matrix 4D volume phased array transducer (4Vc-D). Echo sets were obtained using subcostal bi-plane (**a**) and parasternal short-axis (**b**) views.

**Figure 2 diagnostics-13-01584-f002:**
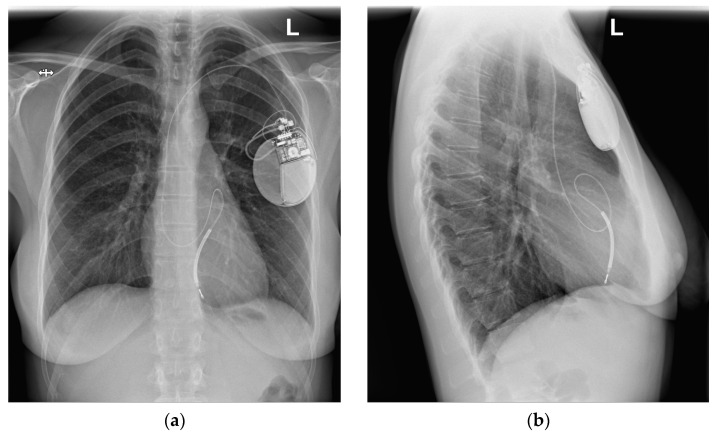
Chest radiograph (CXR) with the transvenous ICD system, location, and lead course in 2021, during the hospitalization, before transvenous lead extraction: (**a**) anteroposterior (AP) view; (**b**) lateral view. The atypical course of the ICD lead can be seen, with the lead body folding into the pulmonary artery and its valve.

**Figure 3 diagnostics-13-01584-f003:**
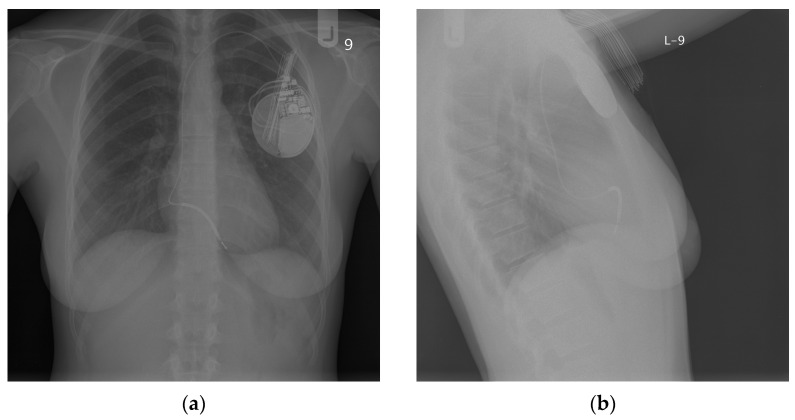
The CXR was performed after the ICD implantation in 2010. The image shows the transvenous ICD system, the initial lead location, and the lead course: (**a**) AP view; (**b**) lateral view.

**Figure 4 diagnostics-13-01584-f004:**
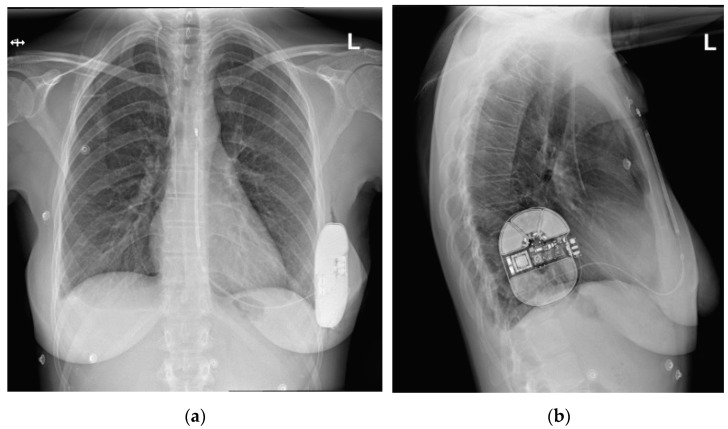
CXR images with the subcutaneous ICD system: (**a**) AP view; (**b**) lateral view. The previous transvenous ICD system was extracted prior to the implantation of the subcutaneous one.

**Table 1 diagnostics-13-01584-t001:** Sample values of parameters during cardiac implantable device control.

Device	Lead Impedance	Sensing (R Wave)	Pacing Threshold	Ventricular Pacing	Follow-Up
Teligen 100 (Boston Scientific)	409 Ohm	20 mV	0.9 V/0.4 ms	0%	No episodes of ventricular arrhythmias, no interventions

**Table 2 diagnostics-13-01584-t002:** Laboratory blood tests of the patient during hospitalization.

Parameter	Result
CRP (mg/L; normal 0–5)	<0.4
PCT (ng/mL; normal 0–0.5)	0.01
D-dimer (ug/L; normal < 500)	302
Hemoglobin (g/L; normal 12–15)	13.1
RBC (T/L; normal 3.8–4.8)	4.57
PLT (G/L; normal 150–410)	292
WBC (G/L; normal 4–10)	9.11

Abbreviations: CRP—C-reactive protein; PCT—procalcitonin; PLT—platelet count; RBC—red blood cell count; WBC—white blood cell count.

## Data Availability

The data presented in this study can be provided upon reasonable request to the corresponding author.

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
