# Peer review of "Application of Novel Technologies in Cardiac Electrotherapy to Prevent Complications"

_diagnostics, 2023, doi:10.3390/diagnostics13091584_

Round 1

Reviewer 1 Report

This case report described a rare case of lead-related complication. Although some similar reports exist, it is considered valuable for clinicians. Some points need to be improved.

1. The mechanism by which the lead enter the body should be described in Discussion. Is this can be explained as a reverse Ratchet phenomenon?

2. Image quality of Figure 3 should be improved.

3. Photographs of the extracted ICD lead should be included if exist.

Author Response

Dear Reviewer,

The authors would like to thank the Reviewer for all the professional and valuable comments. The Reviewer introduced further essential new aspects and allowed the authors to approach their data with ample criticism. Thanks to all the comments we were able to improve the quality of our article. The authors tried to answer the suggestions with due attention and diligence to the limit of their study data. We hope that we have met the Reviewers expectations regarding the revision of our article.

  • The mechanism by which the lead enters the body should be described in the Discussion. Is this can be explained as a reverse Ratchet phenomenon?

    We are grateful for this valuable comment. Indeed, in the literature, macro-dislocation lead-dysfunctioning syndrome, including Twiddler, Reel, and Ratchet syndromes, are casuistically described as a possible explanation of migrated lead; however, in our particular patient, it is difficult to conclude precisely whether a reverse Ratchet phenomenon was in charge of the electrode’s dislocation, as the position of the impulse generator was not altered either the retraction of electrodes was observed or its circling around the pocket. Most probably the suture on the sleeve was not tight enough so the electrode slipped out of the right ventricle. Thanks to the Reviewer’s important comment, new information has been added to the Discussion (lines 178-198) and marked in red, leaving the issue to the further Reviewer's decision.

  • The image quality of Figure 3 should be improved.

    Thank you for that valuable comment. The quality of Figure 3 has been improved now for the required resolution (3480 pixels width&height - minimum 1000 pixels required; and a resolution of 600 dpi – minimum 300 dpi required), leaving it for further Reviewer acceptance.

  • Photographs of the extracted ICD lead should be included if exist.

    We are thankful for that essential comment. The picture of the extracted ICD lead would undoubtedly enrich the manuscript, but, unfortunately, it was not taken. Anyway, routinely, after each procedure, the removed electrode is thoroughly checked by the operator for its eventual changes and/or damage – so it was in our case. The lead did not have any signs of alteration. The new information has been added to the manuscript (lines: 139-141 ) leaving it for further Reviewer acceptance.

Reviewer 2 Report

The authors present a clinical case of intravenous ICD deployment. A rare but interesting case. I would ask the authors for some clarifications:
- in what percentage of cases is the same case reported in the literature, there is a correlation with the underlying pathology?
- was a venography performed prior to device extraction?
- was the patient remotely monitored? if so, were there any alterations in the transmission signals? 

Author Response

Dear Reviewer,

The authors would like to thank the Reviewer for all the professional and valuable comments. The Reviewer introduced further essential new aspects and allowed the authors to approach their data with ample criticism. Thanks to the suggestions we were able to improve the quality of our article. The authors tried to answer the suggestion with due attention and diligence to the limit of their study data. We hope that we have met the Reviewers expectations regarding the revision of our article.

  1. In what percentage of cases is the same case reported in the literature, there is a correlation with the underlying pathology?

Thank you for this valuable question. So far, late lead dislodgment into the pulmonary artery (LDPA) was found only in a few reports (cited in the Article as 14-18, 22- 29) and, usually due to lead fracture and intrusion of the distal fragment into the venous system. Among the cases with LDPA, we did not find an identical case. In our patient, the defibrillating ICD lead was drawn into the venous system from the pocket, possibly due to the inadequate fixation on the anchoring sleeve. The suture was probably not tight enough, so the electrode slipped out of the right ventricle causing its dislocation and its excessive loop was folded into the pulmonary trunk. The summary of available studies describing LDPA is included in Supplementary Table 1. New information is marked in red (lines: 178-198 ). We hope that we clearly understood the Reviewer’s questions and answered them properly, leaving it for further Reviewer’s opinion.

  1. Was venography performed before device extraction?

We are grateful for this insightful question. Venography could have added some important information, particularly before transvenous re-implantation. In that case, we did not perform venography as it was the S-ICD re-implantation, and the operators did not consider venography necessary, additionally minimalising the dosage of radiation for the patient. We add that critical note in the text (141-144).

  1. Was the patient remotely monitored? If so, were there any alterations in the transmission signals?

    Thank you for these essential and important from the clinical point of view questions. Unfortunately, in our country, remote monitoring is not refunded by the National Health Service, and it was not performed on our patient.  This information has been added to the Discussion (lines: 224-226), leaving it for further Reviewer acceptance.
